# Teachers' Perceptions of the Cultural Capital of Children and Families with Immigrant Backgrounds in Early Childhood Education

Lassi Lavanti [1,2,*], Heidi Harju-Luukkainen [2,3] and Arniika Kuusisto [1]

1 Faculty of Educational Sciences, University of Helsinki, 00014 Helsinki, Finland; arniika.kuusisto@helsinki.fi
2 Teacher Education, University of Jyväskylä, Kokkola University Consortium, 40014 Jyväskylä, Finland; heidi.k.harju-luukkainen@jyu.fi
3 Faculty of Education and Arts, Nord University, 8026 Bodø, Norway
* Correspondence: lassi.lavanti@helsinki.fi

**Abstract:** Finnish society has become increasingly diverse relatively recently, notably during the past few decades. This paper explores the perceptions of early childhood education and care (ECEC) teachers (n = 11) about the cultural capital of those children and families growing up in immigrant background families and their sense of belonging in early childhood settings in Finland. We draw on curriculum frameworks to understand these constructs. The paper utilises Bourdieu's concept of cultural capital to conceptualise the sense of belonging for children and families with an immigrant background in Finnish ECEC. The interviews consisted of two pairs and two group interviews with teachers from four ECEC centres, each including two to four ECEC teachers. The data analysis is based on a constructivist grounded theory (CGT) informed content analysis. This approach shows how ECEC teachers' pedagogical practices guide immigrant families and children in Finland towards a national identity. The findings indicate that play and language learning facilitated the development of cultural capital. Nonetheless, it is crucial to investigate the family viewpoint in the future.

**Keywords:** immigrant children; belonging; national identity; early childhood education and care; culture capital

## 1. Introduction

Due to the war in Ukraine, Europe has experienced the most significant war-related immigration since the Second World War [1]. Furthermore, migration between countries and continents is a growing social phenomenon. Naturally, this also affects children and families with immigrant backgrounds in early childhood education and care (ECEC) [2]. Finnish society has experienced a recent increase in diversity, particularly over the last few decades. A systematic review has confirmed that changes in behaviours and attitudes of individuals with an immigration background within an intercultural context can significantly impact their school adjustment and academic achievement [3]. ECEC may also be where immigrant families have their first contact with the new society. There is a lack of research in the field of ECEC in Finland on how to support families with an immigrant background [4–7]. It is, therefore, crucial to investigate how ECEC promotes family belonging and strengthens national identity [8].

Various statistical measures can describe this diversity, but each has its limitations. For example, indicators might include using multiple home languages, adherence to various worldviews, a lack of formal affiliation but active participation, or vice versa. As for languages, by the end of 2020, over 432,800 individuals had a first language other than the national languages of Finnish, Swedish or Sami [9]. Meanwhile, the proportion of the population with an immigrant background among the 5.5 million residents was 8%, equivalent to 444,031 individuals in 2020.

However, there is a significant geographical variation in societal diversity. At the close of 2019, more than half (209,108) of individuals with international backgrounds (423,494) were residents of the Helsinki metropolitan area. Importantly, the concentration of immigrant language and cultural groups is particularly evident among those with a Somali background, 79% of whom lived in the Helsinki Metropolitan Area, followed by those with an Indian background (70%). Regarding age groups, among children in the ECEC age range (0–6 years) in Finland, 11% have an immigrant background, with an even higher percentage in the capital, Helsinki area, where one-quarter of children in this age group have immigrant backgrounds. The largest immigrant groups in Finland comprise individuals who relocated from Russia or Estonia, with Iraq being the subsequent largest group. The average age of the population with migrant background is 33.8 years, in contrast to 43.9 years for the remaining population. About 72,000 individuals born to residents with a migrant background were born in Finland, with an average age of 11.1. This implies that every sixth resident, who statistically has a background other than Finnish, was born in Finland [10]. There is no established framework for categorising immigrant background in Finland. However, it may be determined on the basis of the native language reported [11]. In the ECEC centres where the study was conducted, only data regarding the language backgrounds of the children were collected. Therefore, in this paper, the term 'immigrant background' refers to anyone who has moved to Finland, was born in Finland but has parents with an immigrant background, or speaks a language other than Finnish, Swedish or Sami at home. However, we recognise the influence of intergenerational variance in family history and how individual experiences may differ.

This paper originates from a collaborative project with Australian colleagues and draws on a mutual theoretical framework [12]. This study examines how ECEC teachers construct 'belonging' for families with an immigrant background in early childhood settings in Finland. To achieve this, the paper first draws on curricular frameworks to understand the constructs of belonging in Finnish ECEC. Then, the theoretical framework focuses on cultural capital as defined by Pierre Bourdieu [13]. The study then analyses a group of ECEC teachers (n = 11) using semi-structured interviews from different ECEC centres in Helsinki, using a constructivist grounded theory (CGT) based content analysis [14]. The research question of this study is:

How do ECEC teachers guide families and children in building their cultural capital through teaching practices?

## 2. Literature Review

A sense of belonging is everyone's experience of a secure environment to feel included in something [15]. As Yuval-Davis [16] notes, this also pertains to the emotional connection to a community and feeling at home. In turn, experiencing a sense of belonging can enhance an individual's national identity [17]. A personal sense of belonging differs among individuals, while the politics of belonging encompasses every policy and political document that affects individuals' sense of belonging. Therefore, this section of the paper will examine the policy documents that guide ECEC teachers in Finland.

### 2.1. Finnish ECEC Curriculum

Several policy documents control the provision of ECEC in Finland. The Ministry of Education and Culture is responsible for ECEC at the national level, and the Finnish National Agency of Education is the expert agency for ECEC. The acts are designed by the Ministry of Education and Culture, and tools for their implementation are developed by the Finnish National Agency of Education. Additionally, there are several international, national, and local policy documents that govern ECEC in Finland. At the international level, guidance is provided by the European Commission (1996), the UN (1989, 2006), and the United Nations Educational, Scientific, and Cultural Organization (UNESCO) (1994). Furthermore, the content of ECEC is guided by the national curriculum for ECEC (ages 1–5) [18] and the national curriculum for preschool education (ages 6) [19]. The Finnish

Early Childhood Education Act [20] also stipulates the required child-teacher ratios and the maximum number of pupils per class (12 toddlers or 21 for 3 to 5-year-olds). Furthermore, additional Acts and policy documents guide ECEC settings, although their impact on everyday pedagogical work is relatively small. The Finnish ECEC working teams are multi-professional, comprising professionals with various qualifications. The teams comprise at least one teacher with an academic bachelor's degree and two assistant teachers with a lower educational degree (see [21,22] for more detail). Play and learning through play are essential concepts in the Finnish ECEC curriculum. According to Kangas and Harju-Luukkainen [22], the foundation of the Finnish ECEC, as well as the role of play in the education system, is based on the pedagogy of Fröbel and Pestalozzi regarding children's play and work [23]. The materials, toys, and environment provide children with active learning opportunities through self-directed activities or free play, with teachers only intervening to set the environment and manage unacceptable behaviour [24,25]. Therefore, play is a routine learning activity in Finland's classrooms and outdoor settings [26]. In the Finnish curriculum for ECEC, play is essential for learning, and it is supported with a systematic and goal-oriented approach to scaffold children to engage in learning opportunities. Therefore, teachers must ensure the necessary conditions for play, provide playful guidance and support, and guarantee that all children can play with their peers according to their skills and abilities. In Finland, play is regarded as an attitude, an approach, and a worldview. (see [22] for more detail). A theoretical framework of playful learning and pedagogy has also been created in Finland [22], where playful learning and pedagogy are seen as dialogic and dynamic modalities between free time and set learning, seriousness and fun, care and independent initiatives, togetherness and individual efforts.

### 2.2. Bourdieu's Cultural Capital

Bourdieu [13] uses the concept of capital for theoretical purposes. Capital is typically categorised as economic, social, cultural, or symbolic. Within this study, we apply Bourdieu's notion of cultural capital to the construction of belonging experienced by families of migrant backgrounds in the Finnish ECEC system. Unlike social and economic capital, cultural capital presents as disguised legitimacy of competence. Cultural capital can exist in three forms: embodied, objectified, and institutionalised.

Embodied cultural capital involves the process of "inculcation and assimilation", which takes time to accumulate [13] (p. 18). Bourdieu's concept of embodied cultural capital requires personal investment but is ultimately worth the investment. However, if invested in early ECEC, it has the potential to increase returns to society as the education market becomes more accessible to individuals. Objectified cultural capital encompasses various aspects, including material objects such as texts or paintings. As physical material objects and devices, this type of cultural capital is transferable in its materiality. However, it requires financial capital, such as from a school, to procure novels in various languages.

It should be noted, however, that this type of cultural capital is not only transferrable but also necessitates the assimilation and use of materials. Institutional cultural capital pertains to items disseminated by the state, such as admission to education and degrees. Traditionally, education has provided individuals with a culturally accepted qualification as part of society.

Bourdieu's concept of cultural capital has been criticised for its various interpretations and applications in research [27–29]. Nevertheless, it has proven to be a valuable theory for investigating the concept of cultural capital in education and has motivated several studies [12,30]. Studies that have employed the concept of cultural capital have demonstrated its positive impact on academic performance. Therefore, supporting cultural capital is crucial for reducing educational segregation [31]. Kosunen et al. [30] describe private supplementary tutoring as a typical way economic capital transforms into embodied capital, ultimately resulting in institutional capital. In Finland, individuals with an immigrant background tend to possess a lower educational level than individuals with a Finnish

background [32]. Research by Fanning et al. [33] emphasises the significant role of cultural capital in facilitating integration.

The study by Xu and Hampden-Thompson [31] highlights the significance of the interplay between micro and macro levels of cultural capital. Cultural capital possessed by individuals can be transferred to the macro level. In addition, at the macro level, cultural capital is associated with a nation's welfare system. According to Bourdieu [13], investing in cultural capital at the macro level is the most beneficial way to invest in education, as it can be converted into economic capital at the micro level through education. When children with immigrant backgrounds enter a new ECEC space, devaluing their cultural capital, such as language skills and relationships with peers, can result in a lack of belonging [34]. This paper has opted to utilise Bourdieu's notion of cultural capital despite its criticisms.

## 3. Data

The data for the study were collected through four group interviews, each with two to four ECEC teachers, who represented four ECEC centres (n = 11 in total) and had several years of experience working in the ECEC field. The municipality responsible for managing ECEC in the area granted research ethics permission. The data collection period took place in December 2021, and due to the pandemic, the interviews were conducted digitally via Teams. The interviews lasted between 30 and 50 min each and were recorded and transcribed verbatim. On average, they lasted 44 min (see Table 1).

**Table 1.** Data Collection Information.

| No. of Participants | 11 |
| --- | --- |
| Average length of time per interview | 44 min avg. |
| Participants | 11 ECEC Teachers |
| Type of interview (pair, group)/focus groups | 2 × pair |
| | 2 × group |
| Mode of interviews | Online |

The qualitative research employed semi-structured pair and group interviews to investigate the ECEC curriculum's effect on families' sense of belonging and how teachers facilitate the cultural capital of families and children. Semi-structured interviews were favoured as they allowed for flexible data to be collected to address the research questions [35]. Pair and group interviews were used to gather diverse viewpoints [36]. The researchers formulated open-ended questions for this paper based on themes developed in previous related studies [12].

## 4. Method

The data were analysed using a constructivist grounded theory (CGT)-informed content analysis method [14], following the data-driven and inductive approach. Grounded theory (GT) is a methodology that enables researchers to approach data with various strategies for constructing theory using a qualitative orientation [37]. CGT is an extension for the GT. The CGT extends GT, where the researcher seeks to comprehend the phenomenon through data to analyse existing research and theories, unlike GT, where theories are discovered in data [38]. Our process followed Charmaz's [6] CGT, specifically utilising the structured coding process of open and axial coding and continuously reflecting on Bourdieu's [13] concept of cultural capital and empirical research throughout the data analysis process. (Lavanti, Kuusisto & Harju-Luukkainen, forthcoming.) The coding categories were not pre-defined but instead derived from the data. The analysis used the Atlas.ti program due to its proficiency in data management.

The codes extracted from the data were subsequently grouped into thematic categories [14], as also observed by Elo and Kyngäs [39]. The inductive approach allowed for an open data analysis, aiming to conclude directly from the data [40]. The inductive

analysis involved open coding, category creation, and abstraction [39]. An example of the coding process is presented in Table 2.

**Table 2.** Examples of coding: open coding, category creation and abstraction.

| | |
|---|---|
| Continuously reflection on cultural capital | Open coding: <br> Greetings and learning about family habits Christmas in Finland <br> Family celebrations <br> Organising celebrations in ECEC <br> Making everyone feel welcome at the celebrations <br> Outdoor activities in all weather conditions <br> The right equipment for the weather <br> Reporting on what has been done during the day |
| | Category creation: <br> Celebrations in ECEC <br> How families work <br> Families celebrations <br> Outdoor activities in ECEC <br> ECEC practices in Finland |
| | Abstraction: <br> Families' ways of doing things (How families work, Family celebrations) <br> Different ways of doing things in ECEC (Celebrations in ECEC, Outdoor activities in ECEC, ECEC practices in Finland) |

We used our main concepts related to the research problem and its sub-questions to identify conceptual headings that guide the coding, which was then transferred onto Atlas.ti. During the coding phase, the transcribed data were read, and section headings were noted. These headings were then transferred, from which data-driven codes were developed and used to group headings into categories. The categories were subsequently sorted to identify key themes within the transcribed data. Similar categories were merged during this process. The process of grouping allowed us to establish preliminary categories, which we then consolidated by merging similar categories to identify the main themes from the data. Following this, we embarked on an abstraction phase, as described by Elo and Kyngäs [39]. It involves creating categories that align with the research focus and developing them into specific sub-categories. In our analysis, we initially extracted categories into subcategories. We subsequently identified and connected these subcategories, further unifying them under three forms of cultural capital.

## 5. Findings

The purpose of this research is to examine how ECEC teachers influence families and children's cultural capital through their practices. The analysis of our findings is based on Bourdieu's [13] cultural capital framework, and the results are divided according to the embodied, objectified, and institutionalised forms of cultural capital described in more detail above. The sub-categories are listed in Table 3.

**Table 3.** The sub-categories.

| Embodied | Objectified | Institutionalised |
|:---:|:---:|:---:|
| families' ways of doing things | use of diverse languages in ECEC | ECEC activities |
| language learning | various types of homes | new education system approaches taken towards what and how ECEC is celebrated |
| different ways of doing things in ECEC | various kinds of games played in relation to diverse habits | |

*5.1. Embodied*

We outline how cultural capital is reflected within ECEC based on our analysis. The embodiment of capital was found to consist of families' ways of doing things, language learning and the various ways of doing things in ECEC. Families' practices were seen as noticeable in ECEC through mealtime scenarios wherein children adhere to a distinct diet, unlike most ECEC children. ECEC aims to discuss objectively with children the differences between families and individuals, emphasising the uniqueness of each family's practices. Furthermore, families are provided with an opportunity at the beginning to inform ECEC about their ways of doing things. ECEC strives to ensure that every individual is allocated the space to carry out their practices in the most fitting way.

> "We do the family photos at the beginning of the autumn season and then the families can tell us about their family's celebrations, what they do at home and who is in the family, with pictures or words or something else". (G1T1) (G1T1 = Group 1 Teacher 1)

> "But also, you are allowed to be yourself. And so, we create the opportunity for that". (G4T2)

> "We have the home language of each child's parents, the kind of greetings, flags, and maps. We look at where each person has come from, and then if they naturally come up in conversation, we talk about it and talk". (G2T1)

> "Enabling equality, so that they feel that they are getting something of Finnish culture from this environment. But also that they can give their own culture and be their selves. And that we create the opportunity for that. Just an example of not always making light-faced angels at Christmas, but also dark-faced ones, to give a very small but concrete example". (G4T2)

ECEC aims to facilitate language acquisition for societal integration through diverse methods. One aim is to engage parents in Finnish language discussion. For parents who are fluent in Finnish, staff can aid language learning by conversing with them in routine circumstances. Visual aids are used to support children's language development. Visual aids support children's language development, and efforts are made to strengthen their native language to make it easier for them to learn a foreign language. Children may occasionally use their mother tongue, for instance, during playtime with peers who also speak it. The children's identity is reinforced by inscribing their native languages on the walls of the ECEC centre, thus making all languages spoken by the children in the centre visible in their homes. Learning English via a range of social media channels has presented challenges in acquiring the mother tongue and learning Finnish.

> "A few years ago, the children spoke the basic Finnish they knew and then it built up over time. But now it's English that has come, and the children do speak it to each other, regardless of their home language. This is problematic at the moment because the children are not really learning any language properly". (G2T2)

> "The Finnish language is important for them to learn, so that they can communicate with their fellow countrymen who don't speak the same language they speak at home. And then those skills of making friends and working in a group, because often when the language is missing, it is very difficult to work in the group". (G3T2)

The varied approaches to ECEC may create confusion among certain families. In ECEC, these values are demonstrated through activities such as play. Including Christmas in such education and care often leads to discussions between families and the institutions themselves, as this holiday is intertwined with cultural heritage and religion. Furthermore, the Finnish emphasis on gender equality and neutrality has also been a topic of debate as some personnel report conflicts arising with families.

"Even the children have the question, "Oh, why can't he put an elf hat on his head, or why can't he sing Christmas songs?" (G1T1)

"And there's just a lot of discussion with parents about what Christmas is, and it's more related to the Finnish cultural heritage". (G1T2)

"Everyone can play with everyone and we don't have boys and girls' clothes and toys. So there is a discussion about them from behind the culture". (G1T3)

On the other hand, early childhood education aims to provide support to families, helping them acclimatise to diverse ways of doing things. This is achieved through discussing topics that may cause concern and ensuring families are fully informed about Finnish weather and equipped with appropriate attire for different weather conditions. This is achieved through discussing topics that may cause concern and ensuring families are fully informed about Finnish weather and equipped with appropriate attire for different weather conditions. Additionally, assistance with necessary paperwork is offered.

### 5.2. Objectified

The objectified form appeared in different ways in ECEC. These encompassed the use of diverse languages in ECEC, varying forms of homes, and various types of games in relation to diverse habits. The use of diverse languages in ECEC teaching settings took on various forms. In ECEC, electronic devices are used to play videos, films, animations, songs, or read books in the chosen language, thus supporting the child's home language. Children and families are encouraged to visit libraries and borrow literature or music in their language. During various early childhood education and care ECEC celebrations, children can sing a song or play a game in their native language.

"Many times, we have done it in such a way that if there has been a celebration, we have been able to use songs or materials, for example, a Vietnamese dragon game, Arabic songs or African songs". (G2T1)

"We still have the Lukulumo app for this year's preschoolers. It has stories in several languages, so you can listen to the same story or the same book in Finnish and then in your language, so there are translations. We also familiarise ourselves with each other's languages so that we don't speak them mindlessly, but we practice some songs and then some stories or greetings and practice how to say some words in your language". (G1T3)

"Last year, we listened to children's songs in different languages. The older ones like to try to learn some of the words in their different languages. They have a lot of fun trying to pronounce them themselves". (G3T2)

Various types of homes in ECEC are explored as much as possible. Parents have been requested to share information about their culture with the ECEC if there is something different in it from the mainstream Finnish culture. Additionally, ECEC has tried to visit the homes of children from different cultures. This has allowed for the observation of varied types of homes while demonstrating respect for cultural differences. The pupils have furthermore been granted the chance to bring along personal belongings from their homes, serving to consolidate and augment their understanding of their culture through these items.

"A lot of people invite us to their homes, so we visited different families' homes". (G1T3)

"All sorts of things have happened over the years. We have exhibitions where families bring something from their own culture or a children's rights day, a day where they bring something, like an old toy of a parent and stuff like that. We have had more of an exhibition tradition where parents are challenged to participate". (G3T1)

Conflicts arose due to various types of games played in relation to diverse cultures and customs. In ECEC, children engage in a range of role-plays, which may surprise some parents for whom such activities are unfamiliar in their culture. However, these conflicts may help them in comprehending the values that the Finnish ECEC endeavours to uphold.

"A boy dresses up, even in a skirt. Wants to play with something like this, so when the parents see it, they are a bit surprised and question whether boys can wear something like this". (G1T2)

"Yeah, and in the games, you don't get just that much yet, that everyone gets to play with everyone, and we don't have boys and girls clothes and games. So there is a discussion about them from behind the culture: no, this is not for boys, or these dresses or dolls are not for boys. No clothes of this kind or colours of this kind". (G1T3)

*5.3. Institutionalised*

The institutional form comprises three aspects of ECEC activities, encompassing the new education system and the approaches taken towards what and how ECEC is celebrated. The activities in the ECEC In ECEC, the challenges of the new education system are visible in the challenges that families face in adapting to the different ways the Finnish ECEC system operates. These challenges are largely influenced by the families' cultural background. In some countries, academic skills are prioritised more than in others, and different group management methods are employed. However, the more exposure a family has to Finnish ECEC through their children and acquaintances, the more successfully they adjust to it.

"Families with more children, where someone has already gone to school, have the understanding and knowledge. The perception changes pretty quickly when the trust in us comes". (G1T3)

"I have experienced personally, particularly in the preschool, that some families put a lot of emphasis on learning. And it is of course due to the fact that their children go to school much earlier than here in Finland. Pre-primary education here, for example, does not aim to help children learn to read and write. And when you tell parents this, it can be a terrible disappointment because then what are you doing there if you don't teach these things? You have to talk a lot about what the goal of preschool education is here". (G2T2)

The implementation of the new education system in Finland may present challenges. The new education system poses challenges for families in the ECEC realm. In Finland, for instance, a plethora of forms have to be completed whenever a child shifts from preschool to primary school. Thus, ECEC endeavours to assist families in dealing with these diverse custody forms. Also, there are different support systems for children with special needs than the ones families are used to. Additionally, some families may face cultural tensions when engaging with social services. Nonetheless, positive experiences have bolstered trust in the education system.

"Sometimes, of course, when a child has a special need for support and is approached, there may be a conflict: 'You do this and that here in Finland. We need to open up these issues and talk about the fact that we have not really done so and so, but that it is in the child's best interests that the child receives all the support and needs that they require". (G2T1)

"In some cultures, there is a strong fear or idea that in Finland, if a child is in need of support, the social authorities will come and take the children away. This kind of thinking is quite deep-rooted in some cases. Things are talked openly, and we are told what it really means here, and parents may have a completely wrong idea or picture of what a particular support measure means". (G2T2)

ECEC centre's approach to celebrating has developed to be more accommodating to families' requirements. It endeavours to be sensitive towards different cultures and beliefs, incorporating them where feasible. It endeavours to be sensitive towards different cultures and beliefs, incorporating them where feasible. ECEC implements an annual programme featuring a range of cultural and religious celebrations, as well as adjustments to cater for specific dietary needs. Moreover, ECEC listens attentively to families and children about their celebrations. Additionally, celebrations are adapted to avoid offending other cultures and beliefs. In certain celebrations, families can choose to abstain from participating if they consider it incompatible with their cultural and personal beliefs.

> "We have, for example, an annual calendar that looks at and considers different religions and cultures. Then, of course, what is easiest to consider, if there is something to serve, of course, and of course not to have a program that offends any culture, so that at the Christmas party, some Christmas carols, the "Glory to God" verse is not drawn to a close". (G3T1)

> "They may not participate. The ECEC respects their culture, and we can listen to what they need". (G4T3)

> "When there are religious celebrations, we ask what they are, and then the children are allowed to tell us. We are introduced to just what it is and what it means to them and what it involves". (G1T3)

> "I thought very carefully in the religious context that there would not be any kind of conflict related to religion. For example, last week we had a funny situation here when the City of Helsinki had sent us staff a Christmas gift of this kind of concert, streamed, which could be watched online. We turned it on during our coffee break from our big screen, and it was a concert filmed in a church. We couldn't watch it during the coffee break, because we have children from other cultures, so we had to stop it". (G4T1)

## 6. Discussion

In recent years, Finnish society has experienced increased diversity, including a rise in the number of children with an immigrant background attending ECEC facilities [2]. Additionally, the conflict in Ukraine has resulted in a significant migration issue related to the war [1]. Due to these factors, the research question for this study was: How do ECEC teachers guide families and children in building their cultural capital through their teaching practices? To answer the question, the theoretical framework was based on curricular frameworks to understand the constructs of belonging in Finnish ECEC using Bourdieu's [13] concept of cultural capital. Yuval-Davis [16] stated that belonging relates to the sense of community and feeling at home. The Finnish curriculum for ECEC play is essential for learning, and it is supported with a systematic and goal-oriented approach to scaffold children to engage in learning opportunities.

The sense of belonging is a personal experience of a secure environment in which to feel included in something [15]. The findings demonstrate a significant amount of work being undertaken in ECEC to enhance the sense of belonging of children and families. Various initiatives were implemented, including the support of the native language and the inclusion of different languages, cultures, and worldviews. One example of such an initiative involved showcasing the languages spoken within the environment on the wall. In addition, the families' habits were considered in the activities, and Finnish early childhood education methods were explained in as much detail as possible. According to a teacher interviewed, the goal is to be as equal as possible. In addition, promoting cultural capital was closely related to enhancing a sense of belonging and national identity, which will be further explored in this discussion.

According to Bourdieu [13], the concept of embodiment should be invested in early childhood education. This approach has been adopted in ECEC. Families are given opportunities to share their ways of doing things and, where possible, to do something in their

way. This activity enhances their cultural capital regarding this notion. Additionally, the children's native language is reinforced by various methods.

On the other hand, it is easier for the child to learn Finnish and feel a sense of belonging to a community. The interviews further demonstrated the challenging impact of the absence of the Finnish language on the children's ability to function and communicate within the group. Various materialistic means can enhance the objectified form of cultural capital, which is vital because, in ECEC, this has been addressed by using various materials and increasing the accessibility of materials produced in the family's language. On the contrary, it is questionable what effect negative experiences have on families' cultural capital, as we were told by interviewees that challenges can sometimes arise as the Finnish ECEC does not define play in terms of gender. This, in turn, may have led to conflict situations in some families due to their cultural background. Institutionalised cultural capital is cultivated through education supported through various means, such as offering alternative support. Efforts are made to inform families about the Finnish education system and increase awareness. This is important because the Finnish way of education may differ from what some families are used to.

From these premises, three conclusions have been drawn regarding this study and how ECEC teachers can improve the sense of belonging and national identity for families with an immigrant background. When generalising the findings, it is important to consider the study's limitations. The study was conducted in areas known for their diverse populations, and the teachers who worked there had considerable experience with families from various backgrounds. At the same time, this can also be seen as a positive aspect as we sought to investigate how the staff could assist these families. Thus, this constraint could have led to a broader spectrum of averages in the data than we might otherwise have obtained. However, it could also result in assumptions that may not be applicable in practical day-to-day ECEC settings. In our forthcoming research, we plan to extend this viewpoint even more.

These conclusions include the promotion of cultural capital for both the families and children. Firstly, ECEC operates based on several documents. In Finland, the pedagogy of ECEC includes play as an integral part [22], which was reflected in the findings. Play was utilized to cultivate children's cultural capital, such as facilitating play in their native language or engaging in games that were typical for their cultural background. Secondly, the study finds that language learning plays a crucial part in ECEC in supporting cultural capital. ECEC focuses on improving the child's native language and facilitating the acquisition of the Finnish language. This linked early growth activities from the micro to the macro level, as Xu & Hampden-Thompson [31] argue in their study that cultural capital can be transferred from the micro to the macro level. Learning Finnish facilitates degree attainment, according to Bourdieu [13], and can develop economic capital, which can help at the macro level. Thirdly, ECEC teachers utilized various methods to assist families towards stronger cultural capital, sense of belonging and national identity. The subsequent step would be to investigate how families from an immigrant background perceive these activities. This includes determining whether such activities enhance their cultural capital and, consequently, their sense of belonging and national identity.

**Author Contributions:** Conceptualization, L.L., H.H.-L. and A.K.; methodology, L.L. and A.K.; validation, L.L., H.H.-L. and A.K.; formal analysis, L.L.; investigation, L.L.; resources, L.L., H.H.-L. and A.K.; data curation, L.L.; writing—original draft preparation, L.L.; writing—review and editing, L.L., H.H.-L. and A.K.; visualization, L.L.; supervision, H.H.-L. and A.K.; project administration, L.L., H.H.-L. and A.K.; funding acquisition, L.L. and A.K. All authors have read and agreed to the published version of the manuscript.

**Funding:** Open access and language proofing funded by the Academy of Finland research project (grant no. 356905).

**Institutional Review Board Statement:** The study was conducted in accordance with the Declaration of Research Ethics Committee Finland (TENK), and approved by the municipality of Helsinki (15.10.2021).

**Informed Consent Statement:** Informed consent was obtained from all subjects involved in the study.

**Data Availability Statement:** Data is unavailable due to privacy reasons.

**Conflicts of Interest:** The authors declare no conflict of interest.

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
