# Peer review of "Teachers’ Perceptions of the Cultural Capital of Children and Families with Immigrant Backgrounds in Early Childhood Education"

_education, doi:10.3390/educsci13100977_

Round 1

Reviewer 1 Report

- It mentions grounded theory, but has not substantiated and commented on this, as well as its use, with the original sources. This is a point for review.

- Mentions the perception of culture by pupils and families of immigrant origin... In this respect, I believe that there is no adequate theoretical basis: that is to say, I do not see any legislation on this issue at the general and educational level in the country, nor the terminology used in this (pluricultural contexts with multicultural or intercultural political-educational treatments).

- I think that a more correct terminology should be used when referring to pupils who are children of immigrants. Immigrant is any person who, either due to political, economic or other circumstances, decides (or is forced) to move to another country. Therefore, a child is not a 'migrant pupil', but a 'child of migrants'. These questions would call for a greater terminological correction with regard to the issue of cultural diversity derived from any multicultural context that may occur in a national territory.

- The treatment of qualitative data (categories, etc.) seems to me to be correct. However, perhaps it would have been interesting to include a table showing at a glance each of them, which are then broken down into the sub-categories already included.

Reviewer 2 Report

Thank you for the opportunity to review the study that  examine how ECEC 51 teachers construct 'belonging' for families with an immigrant background in Finland.    The study draws heavily on Boudieus notion of cultural capital and uses grounded theoretical method for analysis of interview data. I think this is a very interesting study providing insights to Finlands ECE and beyond. I however, would like to see some methodological details and a more developed qualitative anaytical step.   METHOD - Here I think there is considerable space to expand on the methodological procedure. What analytical steps and how did the authors work with the data to reach saturation in the GT paradigm? For example, how was data transcribed and coded? What codes and themes emerged? Also, what the interplay was between Boudieus concepts and the data. Detailing this analysis is important so that readers can follow the lines between data and theory, which could also inform the results and discussion.   From a developed method, detailing in  the results sections, for exaple by adding more from the interviews or several perspectives to the themes can further the argumentative power of the study and possibly give more justice to the informants as well.    There seem to be impotant details, nuances and tensions in the material. Still be dicussion centers on cultural capital. I think there is space to  include more of the material and voices of the participants making the qualitative approach more nuanced showing more precisely how the actors work toward belonging for the children and families.

The paper is overall well-written and mostly clear. Some formatting errors and missing letters - a thorough proof-reading could be needed.
